# Procedural Reading Comprehension with Attribute-Aware Context Flow

Aida Amini[1], Antoine Bosselut[1,2], Bhavana Dalvi Mishra[2],
Yejin Choi[1,2], and Hannaneh Hajishirzi[1,2]

[1]University of Washington
[2]Allen Institute for AI
{*amini91, antoineb, yejin, hannaneh*}@*cs.washington.edu*
{*bhavanad*}@*allenai.org*

## Abstract

Procedural texts describe processes (e.g., photosynthesis, cooking) through the state changes undergone by the entities (e.g., plant, food) involved in them. In this paper, we introduce an algorithm for procedural reading comprehension by translating the text into a general formalism that represents processes as a sequence of transitions over entity attributes (e.g., location, temperature). Leveraging pre-trained language models, our model obtains entity-aware and attribute-aware representations of the text by jointly predicting entity attributes and their transitions. Our model dynamically obtains contextual encodings of the procedural text, exploiting information that is encoded about previous and current states to predict the transition of a certain attribute, either as a span from the text or from a pre-defined set of classes. Empirical results demonstrate that our model achieves state of the art results on two procedural reading comprehension datasets: PROPARA and NPN-COOKING.

## 1. Introduction

Procedural text describes how entities (e.g., `fuel`, `engine`) and their attributes (e.g., `locations`) change throughout a process (e.g., a scientific process or cooking recipe). Procedural reading comprehension is the task of answering questions about the underlying process described in the text (Figure 1). This task, in turn, requires inferring attributed of the entities in the process, and their transitions, which might only be implicitly mentioned. For instance, in Figure 1, the `creation` of the `mechanical energy` in the `alternator` can be inferred from the second and third sentences.

Full understanding of a procedural text requires capturing the interplay between all the components of the process: the affected entities, their attributes and their transitions. Recent works in understanding procedural text develop domain-specific models for tracking entities in scientific processes [Mishra et al., 2018] or cooking recipes [Bosselut et al., 2018]. More recently, Gupta and Durrett [2019b] leverage pre-trained language models to obtain general entity-aware representations of procedural text, and predict entity transitions from a set of pre-defined classes independent of entity attributes. Pre-defining the set of entity states limits the general applicability of the model,however , as entity attribute values might be arbitrary spans of text. Moreover, entity attributes can be used for tracking entity state transitions. For example, in Figure 1, the location of `fuel` can be effectively inferred from text as `engine` without the explicit mention of the `movement` transition in the first sentence. In addition, the phrase `converted` in the third sentence gives rise to predicting two transition actions of `destruction` of one type of `energy` and `creating` the other type.

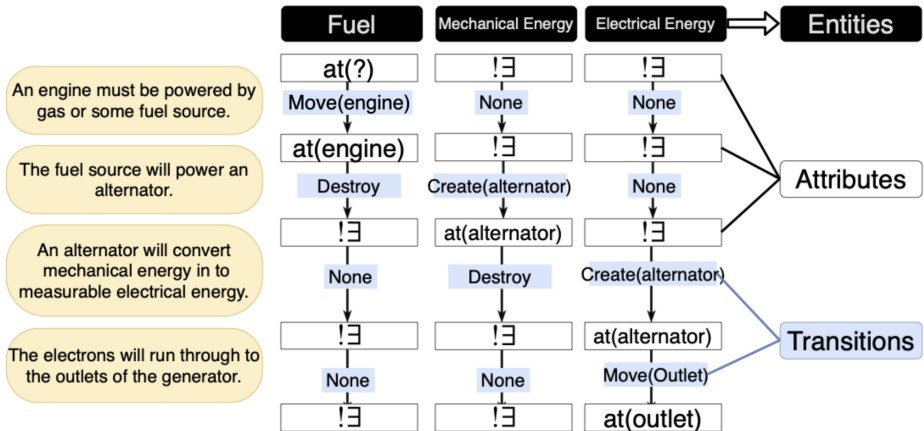

Figure 1: Example of a procedural text and the predicted attributes and transitions for each entity. Procedural reading comprehension is the task of answering questions about the underlying process. Sample questions from PROPARA include: 'What is the process's input?', 'What is the process's output?', 'What is the location of the entity?'.

In this work, we introduce a general formalism to represent procedural text, and develop an end-to-end neural procedural reading comprehension model. Our formalism represents entities, their attributes, and their transitions across time, and the model jointly identifies these attributes and transitions using a dynamic contextual encoding of the text. Our model computes an attribute-aware representation of the procedural text at a certain timestep by leveraging distributions over predicted attribute values, either as a span of text or from a pre-defined set of classes. Then, it pairs this attribute-aware encoding with an entity-aware representation to predict state transitions for entities, thereby capturing the dynamic nature of the entities in the contextual encoding of the process.

Our experiments show that our method achieves state of the art results across multiple tasks evaluated in the PROPARA dataset [Mishra et al., 2018] to track entity attributes and their transitions in scientific processes. Additionally, a simple variant of our model achieves state of the art results in the NPN-COOKING [Bosselut et al., 2018] dataset.

Our contributions are three-fold: (i) we present a general formalism to model procedural text, which can be adapted to different domains, (ii) we develop DYNAPRO, an end to end neural model that jointly and consistently predicts entity attributes and their state transitions, leveraging pre-trained language models, (iii) we show that our model can be adapted to several procedural reading comprehension tasks using the entity-aware and attribute-aware representations, achieving state of art results on several diverse tasks.

## 2. Related Work

Most previous work in reading comprehension [Rajpurkar et al., 2016] focuses on identifying a span of text that answers a given question about a static paragraph. In contrast, this paper focuses on procedural reading comprehension, which inquires about how the states of entities change over time. There are several previous works that focus on understanding procedural text in multiple domains. Mishra et al. [2018] introduced the PROPARA dataset, a collection of procedural texts that describe how entities change during scientific processes (e.g., photosynthesis), which is paired with

questions about several aspects of these processes, such as the entity attributes or state transitions. Bosselut et al. [2018] focus on cooking recipes, which describe instructions on how to change ingredients. Math word problems [Kushman et al., 2014, Hosseini et al., 2017, Amini et al., 2019, Koncel-Kedziorski et al., 2016] describe how the states of entities change throughout mathematical procedures. Narrative question answering [Weston et al., 2015, Kočiský et al., 2018, Lin et al., 2019] inquires about the state of a story over time.

These resources have influenced many models (e.g., EntNet [Henaff et al., 2017], QRN [Seo et al., 2017], MemNet [Weston et al., 2014]) that track entities in narratives. The closest of these works to ours, however, is the line of work focusing on the PROPARA and NPN-COOKING datasets. Bosselut et al. [2018] use an attention-based neural network, the neural process network (NPN), to identify ingredient state transitions in cooking recipes. To track state in scientific processes, models such as Pro-local and Pro-Global [Mishra et al., 2018] first identify locations of entities using an entity recognition approach and use manual rules or global structure of the procedural text to consistently track entities. Tandon et al. [2018] leverage manually defined and KB-driven commonsense constraints to avoid nonsensical transitions when predicting entity states (e.g., a tree doesn't move to a new location). KG-MRC [Das et al., 2019] maintains a knowledge graph of entities over time and identifies entity states by predicting the location spans with respect to each entity using a reading comprehension model. NCET (Gupta and Durrett [2019a]) introduces a neural conditional random field model to maintain the consistency of state predictions. Most recently, $ET_{BERT}$[Gupta and Durrett, 2019b] uses transformers to construct representations of each entity and predict the state transitions from a set of pre-defined classes.

In this paper, we integrate these prior observations, and develop a model that jointly identifies entities, attributes, and transitions over time. Unlike previous work that is designed to address either attributes or transitions, our model benefits from the clues that are implicitly and explicitly mentioned for both entity attributes and transitions. Leveraging both aspects of procedural reading comprehension leads us to a general and adaptive framework for the task, and an accompanying model that achieves state of art on several datasets across different domains.

## 3. Procedural Text Representation

Procedural text consists of a sequence of sentences describing how entities and their attributes change throughout a process. We introduce a general formalism to represent a procedural text:

$$p = (E, A, T), \tag{1}$$

where $E$ is the list of entities participating in the process, $A$ is the list of entity attributes, and $T$ is the list of transitions. We formulate the model's input to be a combination of a query about an entity attribute and a partial context of the procedural text.

**Entities**   are the participants in a process. For example, in the scientific processes in PROPARA, entities might include elements such as `energy`, `fuel`, etc. In the cooking recipe domain, the entities could be ingredients such as `milk`, `flour`, etc. Entities could be given a priori based on the task (e.g., PROPARA) or they could be inferred from the context (e.g., math word problems).

**Attributes**   are entity properties that can change over time. We model attributes as functions `Attribute(e) = val` that assign a value `val` to an attribute of the entity $e$. The entity state at each time is derived by combining all the attribute values of that entity. Attribute values can be

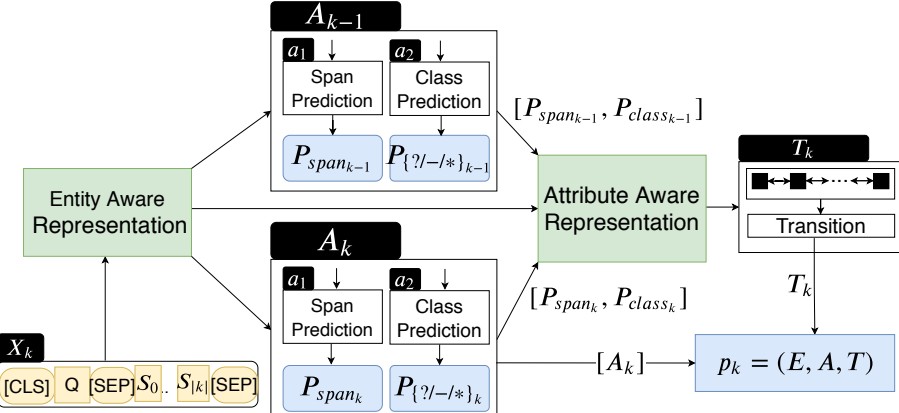

Figure 2: DYNAPRO takes the procedural context $X_k$ as input and predicts attributes $A_{k-1}$, $A_k$ and transitions $T_k$ at each time step $k$. $P_{\{?,-,*\}}$ indicates the probability of the location type among `nowhere`, `unknown`, and `span_of_text` respectively.. The model uses the changes in attribute values from time steps $k-1$ to $k$ to predict transitions.

either spans of text or can be derived from a pre-defined set of classes. For example, in PROPARA, an important attribute of an entity is its `location`, which can be a span of text. The NPN-COOKING dataset introduces several attributes (such as `shape` and `cookedness`) for each ingredient. Example attributes addressing the entities in PROPARA are modeled as follows:

$$\text{exists}(e) = \{\texttt{nowhere}, \texttt{unknown}, \texttt{span\_of\_text}\}$$
$$\text{at\_loc}(e) = \texttt{l} \rightarrow \text{ Assigns the } \texttt{location l} \text{ to entity } \texttt{e}$$

**Transitions**   capture changes in the entity states. More specifically, transitions indicate how entity attributes change over time. We model each transition with an action name and a list of arguments that include the entity and some attribute values. For example, PROPARA consists of four transition types : `Create(e, loc)`, `Destroy(e)`, `None(e)` and `Move(e, loc)`.

## 4. Model

We introduce DYNAPRO, depicted in Figure 2, an end-to-end neural architecture that jointly predicts entity attributes and their transitions. DYNAPRO first obtains the representation of the procedural text corresponding to an entity at each time step (Section 4.2). It then identifies entity attributes for current and previous time steps (Section 4.3) and uses them to develop an attribute-aware representation of the procedural context (Section 4.4). Finally, DYNAPRO uses the entity-aware and attribute-aware representations to predict transitions that happen at that time step (Section 4.5).

### 4.1 Task Reformulation

Given a procedural text $\langle S_0 \ldots S_k \ldots S_T \rangle$ and an entity $e$, DYNAPRO encodes procedural context $X_k$ at each time step $k$ and obtains the entity-aware representation vector $R_k(e)$. The procedural context is formed by concatenating the query containing the entity name, and a fragment of the procedural text. The entity name and the query are included in the procedural context to construct

an entity-aware representation of the context. Since entity attributes are changing throughout the process, we form the context at each step $k$ by truncating the procedural text up to the $k^{th}$ sentence. More formally, the procedural context is defined as:

$$X_k(e) = [cls]Q_e[sep][C_i]S_0 \ldots S_k[sep], \tag{2}$$

where $\langle S_0 \ldots S_k \rangle$ is the fragment of the procedural text up to the $k^{th}$ sentence, $Q_e$ is the entity-aware query (e.g., "Where is $e$?"), $[C_i]$ includes tokens that are reserved for attribute value classes (e.g., nowhere, unknown), and $[cls]$ and $[sep]$ are special tokens to capture sentence representations and separators. Note that the input samples for any paragraph are constructed per sentence and entity. So the complexity of the input data is $|S|*|E|$ per procedural text where $|S|$ and $|E|$ are the average number of sentences and entities in each procedural text respectively.

## 4.2 Entity-aware Representation

DYNAPRO then uses a pre-trained language model to encode the procedural context $X_k(e)$ from Equation 2 and returns the entity-aware representation $R_k(e) = BERT(X_k(e))$, where $BERT(w_i)$ is the embedding of the $i$-th token from the transformer layer. Hereinafter, we will remove the argument $e$ from equations for ease of notation.

## 4.3 Attribute Identification

DYNAPRO identifies attribute values for each entity from the entity-aware representation $R_k(e)$ by jointly predicting attribute values from a pre-defined set of classes or extracting them as a text span.

**Class Prediction**  Certain attribute values are predicted from a set of pre-defined classes. For instance, the existence attribute of an entity is one of $\{$nowhere, unknown, span_of_text$\}$. The distribution $P_{class_k}$ over pre-defined attribute values is predicted from the entity-aware representation $R_k$:

$$P_{class_k} = \text{softmax}(f_{\theta_1}(g(R_k))), \tag{3}$$

where $g$ is a non-linear function, $f$ is a linear function and $\theta_1$ are learnable parameters.

**Span Prediction**  Defining all attribute values a priori limits the general applicability of the procedural text understanding model. Some attribute values are only mentioned within a span_of_text. For example, the full set of locations an entity can be in may be difficult to explicitly pre-define into classes, but may be easily searchable in text. For span prediction, we follow the standard procedure of phrase extraction in reading comprehension [Seo et al., 2016] that predicts two probability distributions over start and end tokens of the span:

$$\begin{aligned} P_{span_k} &= [P_{start_k}, P_{end_k}] \\ P_{start_k} &= \text{softmax}(f_{\theta_2}(g(R_k))) \\ P_{end_k} &= \text{softmax}(f_{\theta_3}(g(R_k))), \end{aligned} \tag{4}$$

where $g$ is a non-linear function, $f$ is a linear function and $\theta_2$ and $\theta_3$ are learnable parameters used to compute the probability distributions over start and end tokens of the span.

In order to capture the transitions of entity attributes, our model captures attributes for time steps $k-1$ and $k$ given a procedural context $X_k$. More specifically, we use Equations 3 and 4 to compute the probability distributions $P_{class_{k-1}}$, $P_{span_{k-1}}$, $P_{class_k}$ and $P_{span_k}$ for both time steps $k$ and $k-1$ at given timestep $k$.

## 4.4 Attribute-aware Representation

For each entity $e$ at each time step $k$, DYNAPRO computes attribute-aware representations $R_{a_k}$ of the context by encoding entities and their attributes using the predicted distributions $P_{span_k}$ and $P_{class_k}$. The intuition is to assign higher weight in the contextual representation to the tokens $w$ corresponding to the attribute value of the entity at time step $k$.

$$R_{a_k} = \sum_{class} (R_k . P_{class_k} \cdot m_{class}) \cdot P_{span_k}(w), \tag{5}$$

where $class \in \{\texttt{nowhere}, \texttt{unknown}, \texttt{span}\}$ are the pre-defined classification of attributes, and $P_{class_k}$ and $P_{span_k}$ denote the probability distributions of attribute values over pre-defined classes and the span of text respectively (as calculated using Equations 3 and 4). $m_{class}$ is a vector that masks out the input tokens that do not correspond to a specific class.

Finally, we model the flow of the context by concatenating attribute-aware representations at time steps $k$ and $k-1$ as:

$$R_{a_{k-1:k}} = [R_{a_k}, R_{a_{k-1}}]. \tag{6}$$

## 4.5 Transition Classification

DYNAPRO predicts attribute transitions from entity-aware and attribute-aware representations. In order to make smooth transition predictions and avoid redundant transitions we include a Bi-LSTM layer before the classification of the transition.

$$\begin{aligned} R_{seq_k} &= \text{LSTM}(h, [R_k, Ra_{k-1:k}]) \\ P_{transition_k} &= \text{softmax}(f_{\theta_4}(g([R_{seq_k}]))), \end{aligned} \tag{7}$$

where $h$ is the hidden vector of sequential layer, $\theta_4$ are learnable parameters and $R_{seq_k}$ is the output of the sequential layer.

## 4.6 Inference and Training

**Training**  Our model is trained end-to-end by optimizing the loss function below:

$$loss_{total} = (loss_{span} + loss_{class})_{k-1} + (loss_{span} + loss_{class})_k + loss_{transition_k} \tag{8}$$

Each loss function is defined as a cross entropy loss. $(loss_{span}, loss_{class})_k$ and $loss_{transition_k}$ are the losses of attribute prediction and the transition prediction modules at time step $k$, respectively.

**Inference**  At each time step $k$, the attributes $A_k$ and transitions $T_k$ are predicted given $P_{span_k}$, $P_{class_k}$, and $P_{transition_k}$. The final output of the model consists of two sets of predictions, the attributes $A_{0...K}$ and transitions $T_{0...K}$ which are combined to track entities throughout a process given a task-specific objective (see §5.3 for further details).

## 5. Experiments and Results

### 5.1 Datasets

We evaluate our model over the PROPARA dataset introduced by [Mishra et al., 2018]. This dataset contains over 400 manually-written paragraphs of scientific process descriptions. Each paragraph

includes average of 4.17 entities and 6 sentences. The vocabulary size of is 2500. The entities are extracted by experts and the transitions are annotated by crowd-workers.

We also evaluate our model on the NPN-COOKING dataset introduced by [Bosselut et al., 2018]. This corpus contains ∼65k cooking recipes. Each recipe comes with a set of ingredients tracked during the process. Training samples are heuristically annotated for attributes and state transitions by string matching, and dev/test samples are annotated by crowd-workers. We randomly sample from the training recipes that have ingredients whose `location` attribute is changed.

## 5.2 Tasks and metrics

We evaluate DYNAPRO on three tasks in PROPARA and one task in NPN-COOKING.

**Document-level Predictions**  This task was introduced by Tandon et al. [2018] and evaluates four different questions per entity and process. These questions involve identifying whether the entity is an (1) input or (2) output of the process, and identifying the (3) moves and (4) conversions of the entity in the process. The final metrics reported for this evaluation are the average precision, recall and F1 score across all four questions.

**Sentence-level Predictions**  This task was introduced by Mishra et al. [2018] and considers three categories of questions. The first, $\mathbf{Cat-1}$, asks if a specific entity is `created/destroyed/moved` in the process. $\mathbf{Cat-2}$ asks the time step at which an entity is `created/destroyed/moved`. Finally, $\mathbf{Cat-3}$ asks about the location where an entity is `created/destroyed/moved`. The evaluation metric calculates the score of all transitions for each question and reports the macro- and micro-average of the scores among three question types.

**Action Dependencies**  This task was recently introduced by Mishra et al. [2019] to check whether the actions predicted by a model influence future events in the procedural paragraph. The metrics reported for this task are the precision, recall, and F1 scores of the dependency links between events averaged over all paragraphs.

**Location Prediction in Recipes**  The task is to identify the location of different entities in cooking recipes. In this domain, the list of attributes are fixed. We evaluate by measuring the change in location [Bosselut et al., 2018] and computing F1 and accuracy in attribute prediction.

## 5.3 Implementation Details

We use the official implementation of $BERT_{base}$ from the `huggingface` library [Wolf et al., 2019]. The learning rate for training is $3e-5$ and the training batch size is 8. The hidden size of the sequential layer is set to 1000 and 200 for class prediction and transition prediction, respectively.

We use the predicted $A_{k-1}$ to initialize the attribute of timestep 0, and at any other timestep, we use the $A_k$ predictions for finding the value of an attribute at timestep $k$. In the sentence level evaluation task introduced in Mishra et al. [2018], the consistency is not required. The inference phase for this task only uses the attribute predictions. For the document-level predictions, we construct the final predictions by favoring the transition predictions. In case of inconsistency (i.e., there is no valid attribute prediction to support the transition), we refer to the attribute value to deterministically infer the transition.

| Model | Sentence-Level | | | | | Document Level | | | Action Dependency | | |
|---|---|---|---|---|---|---|---|---|---|---|---|
| | Cat-1 | Cat-2 | Cat-3 | Ma-Avg | Mi-Avg | P | R | F1 | P | R | F1 |
| ProLocal | 62.7 | 30.5 | 10.4 | 34.5 | 34.0 | **77.4** | 22.9 | 35.3 | 24.7 | 18.0 | 20.8 |
| EntNet | 51.6 | 18.8 | 7.8 | 26.1 | 26.0 | 50.2 | 33.5 | 40.2 | 32.8 | 38.6 | 35.5 |
| QRN | 52.4 | 15.5 | 10.9 | 26.3 | 26.5 | 55.5 | 31.3 | 40.0 | 32.6 | 30.3 | 31.4 |
| ProGlobal | 63.0 | 36.4 | 35.9 | 45.1 | 45.4 | 46.7 | 52.4 | 49.4 | 43.4 | 37.0 | 39.9 |
| KG-MRC | 62.9 | 40.0 | 38.2 | 47.0 | 46.6 | 64.5 | 50.7 | 56.8 | 46.5 | 39.5 | 42.7 |
| NCET | 70.6 | 44.6 | 41.3 | 52.2 | 52.3 | 64.2 | 53.9 | 58.6 | - | - | - |
| NCET + ELMo | **73.7** | 47.1 | 41.0 | 53.9 | 54.0 | 67.1 | **58.5** | 62.5 | 50.4 | 28.6 | 36.5 |
| $ET_{BERT}$ | 73.6 | **52.6** | - | - | - | - | - | - | - | - | - |
| XPAD | - | - | - | - | - | 70.5 | 45.3 | 55.2 | 62.0 | 32.9 | 43.0 |
| DYNAPRO | 72.4 | 49.3 | **44.5** | **55.4** | **55.5** | 75.2 | 58.0 | **65.5** | 64.9 | **32.9** | **43.7** |

Table 1: Results comparing DYNAPRO to prior state of the art methods on sentence-level, document-level and Action Dependency tasks of PROPARA (test set).

To adapt the results of DYNAPRO to identify action dependencies, we postprocess the results using similar heuristics described in the original task. To adapt DYNAPRO to the NPN-COOKING dataset, we use a 243-way classification to predict attributes because the attributes are known apriori.

## 5.4 Results and Analyses

Table 1 and Table 3 compare DYNAPRO with previous models (detailed in Section 2) designed for the PROPARA and NPN-COOKING datasets. As shown in the tables, DYNAPRO outperforms these models in most of the evaluations.

**Document-level Task** We observe the most significant gain (3% absolute in F1) on the document-level task, indicating the model achieves a better global understanding of the procedural text by making joint predictions of entity attributes and transitions. Overall, in most document-level tasks, DYNAPRO predicts transitions with higher precision. In addition, Table 2 shows DYNAPRO's performance on each individual question. The `movement` question achieves the lowest score whereas the `input`/`output` questions are correctly answered more frequently. This pattern is likely due to the fact that `movements` require two span predictions while `create`/`destroy` transitions, which affect the `input`/`output` questions, are less complex.

| Question | P | R | F1 |
|---|---|---|---|
| Inputs | 93.0 | 74.0 | 82.4 |
| Outputs | 81.2 | 91.5 | 86.0 |
| Conversions | 77.5 | 58.3 | 66.5 |
| Move | 53.7 | 47.8 | 50.6 |

Table 2: Precision, Recall and F1 of DY-NAPRO on each question in PROPARA document-level predictions.

**Sentence-level Task** DYNAPRO outperforms the state-of-the-art models on the macro- and micro-average of the three question scores, and gives comparable results to previous work on each individual question type. We note that $ET_{BERT}$ [Gupta and Durrett, 2019b] only predicts actions (`Create, Destroy, Move`), but fails to predict location attributes as spans. DYNAPRO obtains a good performance on the **Cat − 1** and **Cat − 2** questions while also learning to predict answers for questions with more complex structure. We carefully analyze the prediction scores of differ-

ent transitions (`Create`, `Destroy`, `Move`) for each category. In $\mathbf{Cat - 1}$, our model is better than $ET_{BERT}$ for `destroy` transitions, but is worse for `movement` and `create` transitions. In $\mathbf{Cat - 2}$, our model is better than $ET_{BERT}$ for `destroy` and `create` transitions, but is worse than $ET_{BERT}$ for `movement` transitions. Note that our model predicts transitions between time steps by evaluating changes in span predictions, while $ET_{BERT}$ directly predicts the transition class. Directly predicting transitions results in more accurate predictions for `movement` transitions because many of them are explicitly mentioned in the text, but it does not identify the exact locations of `movements`. On the other hand, most `destroy` and `create` transitions are not explicitly mentioned in the text, and using our system to identify changes in entity classes results in more accurate predictions.

**Action Dependency** DYNAPRO outperforms all previous work with F1 score of 43.7. Note that XPAD [Mishra et al., 2019] explicitly favors predicting state changes that result in dependencies across steps. In contrast, DYNAPRO is only optimized to track entities.

**Location Prediction in Recipes** The NPN-COOKING dataset of Bosselut et al. [2018] contains a large number of cooking recipes whose ingredients can be mapped to various attributes. (e.g., `location`, `cookedness`, etc.) A simple variant of DYNAPRO achieves the best performance at predicting the `locations` of ingredients, showcasing the importance of procedural text encoding over time.

| Model | F1 | Accuracy |
|---|---|---|
| NPN | 35.1 | 51.3 |
| KG-MRC | - | 51.6 |
| DYNAPRO | **36.3** | **62.9** |

Table 3: F1 and accuracy on the location prediction task in NPN-COOKING.

### 5.5 Ablation Studies and Analyses

In order to better understand the importance of DYNAPRO's components, we evaluate different variants of DYNAPRO on the document-level task of the PROPARA dataset:

(A.1) **No class prediction** – the model only uses span predictions.
(A.2) **No transition classification** – the model does not include transition classification.
(A.3) **No span prediction** – any token in the span can be predicted using a classifier over the document's words along with the pre-defined class values.
(A.4) **No attribute-aware representation** – the model only considers entity-aware representations in Equation 7 for transition predictions.
(A.5) **CLS instead of attribute-aware representation** – the model uses the `[CLS]` encoding of $R_k$ instead of the attribute-aware representation $R_{a_k}$.
(A.6) **No sequential modeling** – the model removes the sequential smoothing of the predicted transitions by removing the LSTM from Equation 7.
(A.7) **Full procedural input** – uses the full text of the procedure instead of the truncated text $X_k$ at time step $k$.

Table 4 shows that removing each component from DYNAPRO hurts the performance, indicating that joint prediction of attribute spans with classes **(A.1)**, and transitions **(A.2)** are all important in procedural reading comprehension. Importantly, we see that removing span prediction by using a single classifier over a joint vocabulary of document tokens and pre-defined attribute classes **(A.3)** hurts the performance. We also see the importance of combining entity-aware representations with attribute-aware representations that incorporate the flow of context **(A.4)**, and constructing attribute-aware representations in comparison to using a [CLS] token **(A.5)**. The study also shows that using

| Ablation | | F1 |
|---|---|---|
| Full model (DYNAPRO) | | **71.9** |
| (A.1) | No class prediction | 53.8 |
| (A.2) | No transition prediction | 66.3 |
| (A.3) | No span prediction | 55.1 |
| (A.4) | No attribute-aware representation | 69.5 |
| (A.5) | CLS instead of attribute-aware representation | 70.9 |
| (A.6) | No sequential modeling in transition module | 68.8 |
| (A.7) | Full procedural input | 61.0 |

Table 4: Ablation study of different components in DYNAPRO by comparing F1 score on PROPARA **Document Level task** (development set).

a sequential layer for transition modeling can improve final predictions **(A.6)**. Finally, we see the importance of truncating sentences up to a certain time step **(A.7)**, rather than considering the full document. When we increase the context size to include the full document, DYNAPRO does not capture the sequential nature of the procedure and cannot identify the correct attributes at each time step. This shortcoming is partially because the larger document-level context increases the scope of possible span candidates. Over this larger search space, span predictions affect attribute representations more significantly than other predictions (such as pre-defined classes and transitions).

### 5.6 Error Analysis

**Qualitative**  Table 5 shows three types of common mistakes in the final predictions. In the first example, DYNAPRO successfully tracks the `blood` entity while it circulates in the `body`, yet there is a mismatch of what portion of the text it chooses as the span. In the second example, the model correctly predicts the location of `carbon dioxide` as `blood`, but there is not enough external knowledge provided for the model to predict that this entity gets `destroyed` after its removal. In the third example, the model mistakenly predicts the `air and petrol` as a container for the `energy`, but since the changes are explicitly happening to the container, they do not propagate to the `energy`. From these analysis we identified three common patterns in DYNAPRO's errors:

**Incorrect Class Prediction**  Our analyses show that most errors are due to incorrect predictions of `unknowns` vs. `spans` ($\sim 34\%$ of the times `unknown` location should be predicted, DYNAPRO predicts a `span`). Here, we present the confusion matrix of class predictions:

- Gold label as `nowhere`: our system predicts attribute as `nowhere` ($80\%$), `unknown` ($8\%$), and `span of text` ($12\%$).
- Gold label as `unknown`: our system predicts attribute as `unknown` ($56\%$), `nowhere` ($10\%$), and `span of text` ($34\%$).
- Gold label as `span of text`: our system predicts attribute as `span of text` ($67\%$), `unknown` ($18\%$), and `nowhere` ($15\%$).

| # | Sentence | Label | Prediction |
|---|----------|-------|------------|
| 1.1 | Blood enters the right side of your **heart**. | heart | right side of your heart |
| 1.2 | Blood travels to the **lungs**. | lungs | lungs |
| 1.3 | Carbon dioxide is removed from the blood | lungs | lungs |
| 1.4 | Blood returns to left side of your **heart** | heart | left side of your heart |
| 2.1 | **Blood** travels to the lungs | blood | blood |
| 2.2 | Carbon dioxide is removed from the blood. | - | ? |
| 3.1 | Fuel converts to energy when air and petrol mix. | - | air and petrol |
| 3.2 | The car **engine** burns the mix of air and petrol. | engine | air and petrol |
| 3.3 | Hot gas from the burning pushes the **pistons**. | piston | air and petrol |
| 3.4 | The resulting energy powers the **crankshaft**. | crankshaft | crankshaft |

Table 5: Examples of correct and incorrect predictions of DYNAPRO. Entities in the first, second, and third examples are `blood`, `carbon dioxide`, `energy`, respectively.

**Incorrect Span Predictions**    Among predicted spans, $55\%$ of spans are identified correctly. For incorrect span predictions about $9\%$ are due to incorrect span boundary predictions (Predicting `cd` instead of `cd and dvd`) and the rest are due to finding an incorrect phrase.

**Inconsistent Transitions**    We categorize possible inconsistencies in transition predictions into three categories. (The percentages show how many times that inconsistency was observed for all predictions):

- **Creation (2.0%):** When the supporting attribute is predicted to be $non-existence$ or the previous attribute shows that the entity already `exists`.
- **Move (1.5%):** When the predicted attribute is not changed from previous prediction or it refers to a $non-existence$ case.
- **Destroy (1.0%):** When the predicted attribute for the last timestep is $non-existence$.

## 6. Conclusion

We introduce an end-to-end model that benefits from both entity-aware representations and attribute-aware representations to jointly predict attribute values and their transitions for entities in a process. We present a general formalism to model procedural texts, and introduce a model to translate procedural text into that formalism. We show that entity-aware and temporal-aware construction of the input helps yield better entity-aware and attribute-aware representations of the procedural context. Finally, we show that our model can make inferences about state transitions by tracking transitions in attribute values. Our model achieves state of the art results on various tasks over the PROPARA and NPN-COOKING datasets. Future work involves extending our method to automatically identifying entities and their attribute types and adapting to other domains.

## 7. Acknowledgements

This research was supported by ONR N00014- 18-1-2826, DARPA N66001-19-2-403, NSF (IIS1616112, IIS1252835), Allen Distinguished Investigator Award, and Sloan Fellowship. We thank the members of UW NLP group and AI2, and the anonymous reviewers for their insightful feedback.

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
