# OpenReview forum: "Procedural Reading Comprehension with Attribute-Aware Context Flow"
_AKBC.ws/2020/Conference — AKBC 2020_

### Official Review · AnonReviewer2 · 2020-03-16
**The model is reasonable but a bit complex; a simpler baseline seems to outperform it.**

**Rating:** 7
**Confidence:** 4

**Review:**


**Summary**

The paper proposes a model that reads a procedural text, and then tracks the attributes and translations of the participating entities. The tracked information is then used for answering questions about the text.

The key differences from previous works are:

- Both attributes (e.g., at_location(fuel) = engine) and transitions (e.g., Move(destination=outlet)) are tracked. The distribution over attribute values is first predicted based on the contextualized encoding of the entity, and then the transition type is predicted by applying an LSTM over the entity and attribute encodings.

- The attribute values can be either from a closed class or a text span. This is useful for open-class attributes such as "location". However, note that previous work such as Gupta and Durett, 2019 (https://arxiv.org/abs/1904.03518) also considers text spans as possible values.

The approach is evaluated on ProPara and npn-Cooking datasets. The method outperforms the baselines in most categories.

**Pros**

1. The method looks reasonable and empirically performs well (with some caveats: see Cons 1 and 2).

**Cons**

While joint modeling (of attributes and transitions in this case) is intuitively attractive, a strong baseline that does not do joint modeling should be considered more seriously.

1. The proposed joint model is pretty complex compared to the ET_BERT baseline, which simply embeds the formatted text and makes predictions directly. Yet ET_BERT outperforms the proposed model in the two sentence-level tasks. It is unclear if the proposed model would outperform ET_BERT if it is extended to other tasks (which can be done by changing the set of possible prediction targets from {created,moved,destroyed} to, for example, the set of all text spans).

2. According to the ablation study (Table 3), the model does not seem to gain a lot from jointly modeling attributes and transitions (only a 1-2% drop for "no attribute aware representation" and "no transition prediction", whereas the differences in document-level F1 scores between models in Table 1 is much larger).

**Questions**

1. In Section 5.4, the reason why the proposed method lags behind ET_BERT is described as "highly confident decisions that lead to high precision, but lower prediction rate". Would it be possible to give more details? What would be the score if the model is forced to predict on all examples (100% prediction rate)? What are the precision/recall/F1?

2. Table 3 lists a "no class prediction" but not a "no span prediction". What is the F1 when only class predictions is allowed?

3. Contribution (b) in the introduction states that the model "consistently" predicts entity attributes and transitions. What does "consistency" refer to in this context? While the attributes and transitions are modeled jointly, there is no explicitly consistency check between the two.

4. How often do the prediction errors belong to type 1 in Table 4 (technically correct; span boundary mismatch)?

---

### Official Review · AnonReviewer1 · 2020-03-28
**Novel but lacks convincing experimentation**

**Rating:** 7
**Confidence:** 3

**Review:**

The paper introduces an end-to-end model for understanding procedural texts. The novelty of the model lies in jointly identifying entity attributes and transitions of entities as described in the text. The model is evaluated on two standard datasets, ProPara and NPM-Cooking and shows that jointly modeling attributes and transitions helps.

The model itself is quite straightforward; it computes an entity are representation of the text, then predicts a distribution over entity-attributes which leads to an attribute aware representation of text.  Using the attribute representation of the current time step and the previous one, the model predicts the entity-transition. Since the attribute is being inherently predicted, the attribute representation is computed by marginalizing over different attribute values.

I think the main weakness of the paper is in the lack of evaluation and analysis:
1. In Cat-1 and Cat-2 categories in sentence-level experiments in Table 1, the proposed model lacks being previous work [Gupta & Durrett 2019b] which is fine, but does not perform analysis on why this happens. On the contrary, [Gupta & Durrett 2019b] do show that when looked at class-based accuracy, their model struggles in the ``"movement" class. It would be important to know how the current model fares. They also note that the challenge in this dataset is when new sub-entities are formed or entities are referred to with different names. It would be important to see analysis of such kind.

2. On the NPM-Cooking dataset the paper invents a new location-prediction experiment and evaluates their model only on that. This experiment seems quite thin given that the original paper [Bosselut et al., 2018] proposes two tasks, Entity-selection and State-change, both of which the current model should be capable of performing.


Computational complexity --  Since the model computes entity representation at every time-step and for each entity separately, the paper should explicitly point out the computation complexity of computing entity representations as E * T.

Writing / Formatting

1.  Explicit reference to NPN-cooking is never given.
2. The paper contains quite a few spelling mistakes. E.g. t_loc(e)="enginge" in Fig 1.
3. The notation is confusing sometimes. E.g. After Eq. 3. it says "where X0 . . . Xk is the .." which I think should be "where S0 . . . Sk is the "

---

### Official Review · AnonReviewer3 · 2020-03-31
**Simple, yet powerful idea of recasting procedural text understanding task into a query-context reading comprehension task. Model is simple, results are quite compelling.**

**Rating:** 9
**Confidence:** 3

**Review:**



Summary of the paper
=================
This paper addresses the task of understanding procedural text, using a reading comprehension based approach. This work proposes jointly modeling entity attributes and transitions in order to build robust entity and attribute representations. The paper shows that using a modern pretrained language model, this system is able to beat previous approaches on ProPara and a cooking recipe based dataset (NPN-Cooking).

Overall Review
============
The idea of formulating the procedural text understanding task as a reading comprehension task, where the question/query is encoded alongside the context is simple, yet powerful. This technique side-steps the issue of properly encoding which entities and attributes one wants the model to focus on, as well as how to extract representations for these aspects. I also found the derivation of transition representation, stemming for subsequent attribute states to be relatively simple and intuitive. Overall, this paper is generally well written, with mostly clear exposition of ideas. The results are quite strong and compelling.

Pros
====
1. Relatively simple and powerful (re)formulation of the procedural understanding task into a reading comprehension / question-answering formulation.
2. Model design is relatively simple and seems likely to be reproducible.
3. Results are quite strong and compelling,
4. Experimental design seems to be well done, along with interesting ablations and analyses of results

Main points to address
=================

1. My main point to improve is that the paper could have a more clear separation between task (re)formulation and model implementation. Although Section 3 describes the basics of procedural text, it is only while reading the model implementation (Section 4) that the reader is exposed to the specific task re-formulation proposed in this paper.
As far as I know from recent related work, none of the systems formulate the task as:
[“Entity query?”, “S0”, … “Sk”], that is, encoding the actual query in textual form as part of the input to the task. Typically the inputs to the model are simply [“S0”, … , “Sk”] or (recently with Gupta and Durrett, 2019) the task input is [“Entity”, “S0”, … “Sk”].

2. In Section 4.1, the entity representation is described as: Rk(e) = BERT(Xk(e)). This is a bit confusing since it’s unclear what the BERT() operator is exactly. Is this the output representation of the CLS token from the Transformer? Or something else? Please add a clear description of this representation.

3. My understanding is that computing A{k-1} and A{k} at each timestep is done to compute representations for T{k}. However, I think it would have been interesting to see how A{k} for one step compares to A{k-1} of the next step. Presumably they are predicting over the same set of entity attributes at the same time. But the later step has more (“future”) context to look at. Does this improve or degrade the quality of the attribute representation?

4. Typo in Figure 1;  “at_loc(e)=enginge” should be “at_loc(e)=engine”.

5. Regarding the “NPN-Cooking” dataset, Section 5.1 does not reference where the dataset comes from. It took me a while to find the dataset, and realize NPN comes from “Neural Process Networks”. Since the dataset had no name in the original paper, please be more clear about what the dataset is.

6. In Figure 2, there are a couple of confusing labels. On the top attribute representation (A{k-1}), the output is marked with [P(span{k}), P(class{k})]. Is this correct? Perhaps switched with the bottom attribute representation that refers to A{k} ?

7. In Section 4.5, Inference and Training, there is a reference to a term “loss_{state}” which is not defined. Perhaps this was meant to be “loss_{class}” ?

---

### Decision · Program_Chairs · 2020-05-01

**Decision:**

Accept

**Comment:**

This paper proposes a model for reading comprehension of procedural texts that jointly predicts entity attributes and transitions in entity state.  The model achieves state-of-the-art results on multiple datasets.  Reviewers appreciated both the novelty and the simplicity of the proposed model, as well as the strong empirical results.  The initial discussion focused around analysis of errors, ablations, and empirical comparison to prior work.  The responses and revisions addressed these questions and reviewers appreciated the usefulness of the new error analysis and ablations.